# Small Area Estimation of Zone-Level Malnutrition among Children under Five in Ethiopia

**Kindie Fentahun Muchie** [1,2,*] , **Anthony Kibira Wanjoya** [3] **and Samuel Musili Mwalili** [3]

1    Institute for Basic Sciences, Technology and Innovation, Pan African University, Nairobi 62000-00200, Kenya
2    Department of Epidemiology and Biostatistics, Bahir Dar University, Bahir Dar 6000, Ethiopia
3    Department of Statistics and Actuarial Sciences, Jomo Kenyatta University of Agriculture and Technology, Nairobi 62000-00200, Kenya; awanjoya@gmail.com (A.K.W.); samuel.mwalili@gmail.com (S.M.M.)
*    Correspondence: muchie.kindie@students.jkuat.ac.ke

**Abstract:** Child undernutrition is one of the 10 most significant public health problems worldwide. There is a rapidly growing demand to produce reliable estimates at the micro administrative level with small sample sizes. In this research, the authors employed small area estimation techniques to estimate the prevalence of malnutrition at the zonal level among children under five in Ethiopia. The small area estimation concept was sought for by linking the most recent possible survey data and census data in Ethiopia. The results show that there is spatial variation of stunting, wasting and being underweight across the zone level, showing different locations facing different challenges or different extents.

**Keywords:** undernutrition; prevalence; hierarchical Bayesian; spatial analysis; small area estimation; Markov chain Monte Carlo

## 1. Introduction

Malnutrition is defined as an imbalance in the quantity of protein, calories, and other nutrients consumed, usually including either undernutrition or overnutrition. Undernutrition is usually characterised by stunting, wasting, or being underweight. Child undernutrition can have an immediate impact on child mortality and morbidity, or it can have a long-term influence on the labour market and health consequences in adults.

Globally, in 2020, it was estimated that 149 million children under the age of 5 were stunted, 45 million were wasted, and 38.9 million were overweight or obese. Undernutrition among children is a significant public health issue in developing nations, as evidenced by the fact that undernutrition is ranked as the first priority among the world's 10 most important challenges. Ethiopia is one of the countries in the world with the highest rates of childhood undernutrition. Despite significant progress toward eliminating undernutrition in Ethiopia, between 2005 and 2019, the proportion of underweight children decreased from 33% to 21%, the proportion of stunted children decreased from 51% to 37%, and the proportion of wasted children decreased from 12% to 7% [1].

Both the World Health Assembly and the Sustainable Development Goals (SDGs) papers clearly emphasise the necessity of member countries implementing nutrition policies that prioritise maternal and child nutrition [2,3]. Implementation of these ambitious nutritional objectives outlined in the national and global agreements needs to be backed up with an unceasing stream of up-to-date evidence. Actions to eliminate malnutrition are also crucial for reaching the diet-related objectives of the global strategy for women's, children's, and adolescent's health for 2016–2030 [4] and the 2030 agenda for sustainable development [5]. The World Health Organization (WHO) likewise envisions a world free of all types of malnutrition in which all people attain good health and well-being. The WHO collaborates with partners and member countries to achieve universal access to healthy diets and effective nutrition interventions derived from resilient and sustainable food systems,

according to the nutrition strategy of 2016–2025 [6]. The government of Ethiopia has taken several steps toward reducing undernutrition in the country. The recently endorsed 2019 Food and Nutrition Policy aims to achieve an optimal nutritional status throughout the life cycle via coordinated implementation of nutrition-specific and nutrition-sensitive interventions. In addition, through the Seqota Declaration, Ethiopia has committed to ending undernutrition in children under the age of 2 by 2030.

According to worldwide data, many diverse interventions enhance undernutrition outcomes, yet comparable interventions have varying effects in various situations and places [7]. In Ethiopia, several correlations and interventions have also been found to be important for undernutrition outcomes, including food aid and shocks [8,9], maternal nutritional and educational status [10–13], access to educated and trained health workers [14], access to feeding practices, and safe water [15,16]. The prevalence of various forms of undernutrition varies by geography in Ethiopia, showing that different geographic areas confront distinct problems with undernutrition [17].

The national-level Demographic and Health Survey (DHS) is the main source of official statistics in developing countries where there is no vital registration. The DHS data help generate a variety of relevant statistics at the macro level (administrative level and national levels). These days, there is a rapidly growing demand for micro-level statistics. However, the DHS data cannot be utilised directly to generate valid estimates at the micro level because of small sample sizes. Hence, employing small area estimation (SAE) is of paramount importance.

Even though there was a study conducted using 2014 Ethiopian Mini DHS in combination with census data to find small area estimates at the woreda level, it is not appropriate to estimate at the woreda level as the survey data are shifted spatially, where the shifting guarantees that the clusters will not be out of the zonal level [17]. It is also important to use the latest possible data: Ethiopian Mini DHS (EMDHS) 2019 data. In this work, SAE approaches were employed to obtain model-based estimates of the prevalence of malnutrition at the zone level in Ethiopia by linking data from the EMDHS 2019 and the Ethiopian Census 2007.

## 2. Methods and Materials

### 2.1. Study Setting and Design

Ethiopia is organised into four administrative levels: region, zone, woreda, and kebele. Ethiopia's first administrative division is the region, also known as a kilil or, alternatively, a regional state. The regions of Ethiopia are defined by ethno-linguistic areas. Currently, in 2022, there are 11 regions (Afar, Amhara, Oromia, Benishangul-Gumuz, Somali, Gambela, Harari, Sidama, Southern, South West, and Tigray) and 2 independently administrative cities (Addis Ababa and Dire Dawa). Zones are created by subdividing regions. Zones are administrative subdivisions in Ethiopia where DHS shifting guarantees that no survey clusters are outside of the zone. Zones are further subdivided into woredas, and woredas are also subdivided into kebeles. Going back over time within the country, woredas are generally stable administrative entities. Kebeles are the lowest administrative units or divisions of Ethiopia.

This study is a further analysis of secondary data: the Ethiopian Mini DHS (EMDHS) 2019 and the Ethiopian Census 2007. The Ethiopian Census 2007 is the country's third population and housing census and was conducted on 28 May 2007 for all regions except Afar and Somali, which were enumerated on 28 November 2007. The EMDHS 2019 was designed to represent national, urban-rural, and regional estimates of health and demographic outcomes. The samples for the EMDHS 2019 were chosen using stratified and two-stage cluster sampling procedures. Sketch maps were drawn for each of the clusters, and all conventional households were listed.

The EMDHS 2019 was a nationwide survey that included a nationally representative sample of 9150 randomly selected households. In the selected households, all children under the age of five were eligible for measurements of weight and height. The survey was

designed to generate reliable estimates of important indicators for urban and rural areas, for each of the regions, and at the national level in Ethiopia. Figure 1 depicts the clusters included in the 2019 EMDHS.

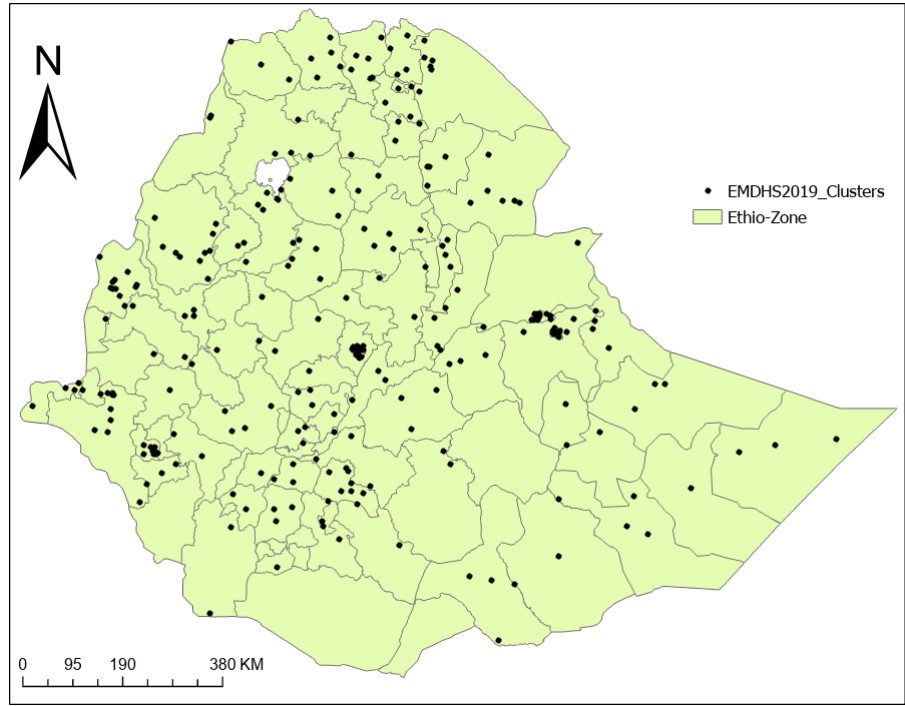

**Figure 1.** Study clusters of the EMDHS 2019.

### 2.2. Data Sources and Procedures

The analysis of the current study was based on the most recent available data: the EMDHS 2019 and the Ethiopian Census 2007. The EMDHS 2019 is a nationally representative cross-sectional household-based sample survey designed to provide information about several health and nutritional indicators in Ethiopia. The principal objective of the EMDHS 2019 was to provide up-to-date regional- and national-level estimates of the indicators. Specifically, the nutritional statuses of children under the age of five was assessed by measuring their weights and heights.

The EMDHS 2019 was carried out with the use of standardised data collection procedures and survey design. The EMDHS 2019 used a stratified cluster sampling technique to choose census enumeration areas (EAs) based on probability proportional to the enumeration area sizes. Following that, a random sample of households within the selected EAs was chosen. Data for the survey were gathered through face-to-face interviews with questionnaires administered to household heads and chosen household members who consented to being interviewed. Data collection for the EMDHS 2019 took place from 21 March to 28 June 2019. The EMDHS 2019 data for the current study were extracted from the Major DHS ((http://dhsprogram.com, accessed on 5 February 2021).

The 2007 Census of Ethiopia is one of the biggest and most recent data sources in Ethiopia, providing information on a wide variety of demographic, socioeconomic, and educational characteristics and the migration statuses of people at a disaggregated level. For the 2007 Census, short- and long-form questionnaires were created for use. The long form was administered to 20% of the randomly chosen households, which covered both housing and population topics. Specifically, the long form was composed of questions about internal migration and geographical characteristics, population characteristics, family and household characteristics, social and demographic characteristics, mortality and fertility, educational and literacy characteristics, disability characteristics, and economic characteristics. The long form was rich in terms of data, with demographic information such as assets, housing characteristics, education, fertility, and mortality.

The short format covered only basic demographics, and 80% of the households received the short format. All the questions in the short form were included in the long form. As a result, the short form was developed to gather information from the whole population. To collect information from individuals, households, and institutions, both the short and long forms were employed. Only the short-form questionnaire was used for the resident foreigners and the homeless. The 2007 population and housing census designated 86,805 enumeration areas (in all regions) with 17,363 urban areas and 69,462 rural areas. The current study's census data was obtained from the Integrated Public Use Microdata Series (IPUMS) (https://international.ipums.org, accessed on 12 May 2021).

All available shape files were collected. We used woreda-level geographic boundary mapping within regions to join data from different sources because woredas are relatively stable government structures, compared with structures below or above the woreda, during political or government changes.

### 2.3. Response Variable

This study applied the recent possible available survey—the EMDHS 2019—unlike the EMDHS 2014, which was used by a previous study [18].

The outcome variables considered were childhood stunting, wasting, and being underweight, which are binary at the individual level. According to WHO criteria, children with height-for-age, weight-for-height, and weight-for-age z-scores of less than $-2$ standard deviations (SDs) were leveled as stunted, wasting, and underweight, respectively. The parameter of interest was estimating the zone level prevalence of stunting, wasting, and being underweight among children under five. Notably, the EMDHS 2019 was not designed to provide zone-level estimates for key maternal or child health indicators, including childhood undernutrition, and therefore this study did not use the SAE technique to estimate the prevalence of stunting, wasting, and being underweight. The study restricted our analysis to children under five, as the EMDHS 2019 collected information on key indicators of child health and development only for those who were born in the five years prior to the survey.

### 2.4. Auxiliary Covariates

In this study, the auxiliary variables were taken from the 2007 Population and Housing Census of Ethiopia. The chosen covariates were at the district level and zonal level. Literacy is widely acknowledged to benefit the individual and society and is associated with a number of positive outcomes for health and nutrition. We considered household characteristics, including education, as an auxiliary variable, as they were used from a study conducted in Ethiopia using the EMDHS 2014 data [18]. As the number of auxiliary variables we can consider should be small in number, we selected a few of the available variables in the census data, namely those which explained the outcome variable more. Hence, we summarised both the woreda- and zone-level summary values (percentages) of the auxiliary variables, including the percent of literacy and percent of access to an improved water supply.

### 2.5. Data Processing and Analysis

The analysis was estimated first by mapping the census data at the woreda-level and zone-level areas. Then, the data sources were mapped by subnational states before processing the data in the analysis. We overlaid the population grid on a current shapefile and then aggregated the population within each area to generate woreda-level and zone-level population estimates for the country. Zone-level estimates of prevalence of stunting, wasting, and being underweight were compiled in the country.

Let $Y_i(A_r)$ be a binary response variable for the $i$th individual in the $r$th area $A_r$, where $i = 1, \ldots, N_r$, $r = 1, \ldots, n$ and $\sum_{r=1}^{n} N_r = N$, with $n$ referring the number of small areas and $N_r$ referring the number of individuals in the $r$th area. We considered $y_i(A_r)$ as taking values one (with probabilities $p_i$) and zero (with probabilities $1 - p_i$), being a realisation of a random variable $Y_i(A_r)$ following a Bernoulli distribution (i.e., $y_i(A_r) \sim ber[p_i(A_r)]$).

For comparison purposes, we also computed the zone-level estimated direct prevalence of malnutrition and its corresponding variance. We used sampling weights to compute the estimated weighted direct prevalence. Similarly, variances were computed using Taylor series linearisation [19] to estimate the variance, considering the sampling weight.

Spatial analysis (both global and local spatial autocorrelation) were used so as to check the importance of including the spatial effect in the modeling. The survey GPS coordinates were combined with the weighted prevalence of stunting, wasting, and being underweight in each of the EMDHS 2019 clusters. As a result, the cluster level weighted prevalence was used to depict the hot and cold spots of clusters. Geographic variation in stunting, wasting, and underweight prevalence among the EMDHS 2019 clusters was identified using spatial analysis [20]. Geographic variation of a significant high prevalence or low prevalence of stunting, wasting, and being underweight was computed for each cluster using Moran's I statistic [20]. Maps depicting the distribution and variations of stunting, wasting, and being underweight throughout the country were constructed.

Regarding the small area estimation, we adapted the Bayesian approach of modeling for its ability to combine information from several sources [21]. The approach also simplifies computation of the measures of accuracy in the SAE, which produces realisations of the posterior distribution of the target quantities [22]. Empirical Bayesian (EB) and hierarchical Bayesian (HB) methods are Bayesian approaches which are more generally applicable in the sense of handling models for binary and count data [23]. The ELL method [24], which is an EB estimation method used by some authors [18,25], assumes a nested error model on the transformed variables [23]. Though the ELL method can handle data from survey and census sources, its nested error modeling nature requires individual-level auxiliary variables, which we could not get from the census data for the individuals in the survey. Under the HB framework, there are a number of developed models for discrete outcome variables [26–28].

Accordingly, we used a spatial hierarchical Bayesian small area model for the binary response variable [28], which enabled us to use area-level auxiliary data from the census and individual-level data from the survey. We considered zone-level classification as small area two, which we wanted to estimate, whereas woreda-level classification was classified as small area one. Furthermore, 72 knot points within two resolutions were considered in the spatial dimension reduction. Further details of the modeling can be found in a published study elsewhere [28].

Weakly informative priors were considered for the model parameters. All the priors used being proper guaranteed that the appropriateness of the posterior distribution. Markov chain Monte Carlo (MCMC) simulation was used to generate posterior samples from the conditional distributions of the parameters of the model. Inspection of the plots (trace plots, density plots, and autocorrelation plots) and formal tests (Geweke's test) were considered in order to check the convergence of the simulated sequences in the models.

Measures of precision play a crucial role in small area estimation. Consider $R$ as the number of MCMC samples after removing the burnin period followed by thinning. By the ergodic theorem for the Markov chains [29], $\hat{p}_i$ converges to $E(p_i|y)$ and $\hat{V}(p_i|y)$ to $V(p_i|y)$ as $R \longrightarrow \infty$. Checks of the convergence of the MCMC were used to guarantee the ergodic theorem. Hence, the estimate of $p_i$ and its corresponding posterior variance for the $i$th area are obtained directly from the predictive distribution of $y(A_k^*)$ accordingly:

$$\hat{p}_i \approx \frac{1}{R} \sum_{k=1}^{R} \hat{p}_i^{(k)} = \hat{p}_i^{(\cdot)}$$

and

$$V(\hat{p}_i) \approx \frac{1}{R-1} \sum_{k=1}^{R} \left( \hat{p}_i^{(k)} - \hat{p}_i^{(\cdot)} \right)^2$$

Benchmarking is important in that the model-based estimators do not benchmark against the direct survey estimate for large areas [30]. To avoid possible overshrinkage and

model misspecification, the model-based HB estimates $\hat{p}_i$ were benchmarked so that the benchmarked HB (BHB) estimates added up to the direct large area (regional level in this study case) estimate. The posterior mean squared error (PMSE) was used to measure the variability of the BHB estimators. The PMSE is the sum of a bias correction term and the posterior variance.

We take $\hat{p}_i^{BHB}$ as the benchmarked HB (BHB) estimator of $p_i$ such that $\hat{p}_i^{BHB}$ is a function of the HB estimators $\hat{p}_i^{HB}$ (i.e., $\hat{p}_i^{BHB} = f(\hat{p}_1^{HB}, \ldots, \hat{p}_n^{HB})$) for some function $f(\cdot)$, satisfying the benchmark property [30] $\sum_{i=1}^{n} \hat{p}_i^{BHB} = \sum_{i=1}^{n} \hat{p}_i^{Direct}$., where $i = 1, \ldots, n$ and $\hat{p}_i^{Direct}$ is the direct survey estimator. Hence, the BHB estimator can be obtained as follows:

$$\hat{p}_i^{BHB} = \hat{p}_i^{HB} \frac{\sum_{k=1}^{n} \hat{p}_k^{Direct}}{\sum_{k=1}^{n} \hat{p}_k^{HB}}$$

To obtain a measure of variability associated with the BHB estimator $\hat{p}_i^{BHB}$, we use the following posterior mean squared error (PMSE):

$$\text{PMSE}\left(\hat{p}_i^{BHB}\right) = \left(\hat{p}_i^{BHB} - \hat{p}_i^{HB}\right)^2 + V\left(\hat{p}_i^{HB}\right).$$

Thus, the PMSE of $\hat{p}_i^{BHB}$ is simply the sum of the posterior variance $V(p_i \mid y)$ and a bias correction term $\left(\hat{p}_i^{BHB} - \hat{p}_i^{HB}\right)^2$.

## 3. Results

### 3.1. Data Description

Both census and survey data were considered in this study. We obtained 10% of the census 2007 data, with 7,434,086 individuals covering all 720 woredas at the time of the census. Out of these, 6,132,270 had short-form data. Accordingly, the remaining 1,301,816 participants with the long-form data type were considered to compute the woreda- and zone-level auxiliary variables for our analysis. We summarised the woreda- and zone-level weighted averages and percentages of the auxiliary variables.

From the EMDHS 2019, 4552 children under 5 with complete information of their anthropometric measurements (height, weight, and age) were considered in the analysis. Furthermore, the global positioning system (GPS) data (position of enumeration areas) of the EMDHS 2019 and shapefiles were overlaid to demonstrate the estimates visually.

### 3.2. Direct Estimates of Malnutrition

The weighted prevalence of malnutrition was computed as a direct estimate at the woreda level and the estimates are given in Table 1. The corresponding 95% confidence intervals were computed using variance estimates from Taylor series linearisation.

**Table 1.** Weighted direct prevalence of indicators of malnutrition among children under 5 in Ethiopia in 2019 (n = 4552).

| Indicator | Prevalence | SE | 95% CI |
|---|---|---|---|
| Stunted | 38.9 | 1.29 | [36.38, 41.45] |
| Wasting | 22.5 | 1.36 | [19.81, 25.14] |
| Underweight | 6.9 | 0.53 | [5.82, 7.89] |

From Figure 2, we can see the number of clusters included and the direct estimate for the weighted prevalence of malnutrition. Specifically, there were a few zones with zero clusters included in the survey (Figure 2a), and hence direct estimates for these zones could not be found (Figure 2c–e). Hence, small area estimation is of paramount importance to obtaining estimates for zonal level administrative classification. That aside, it also important to check whether taking the spatial effect into account in the small area estimation process improves the estimates or not. Spatial autocorrelation analysis and spatial pattern analysis helped us to check this.

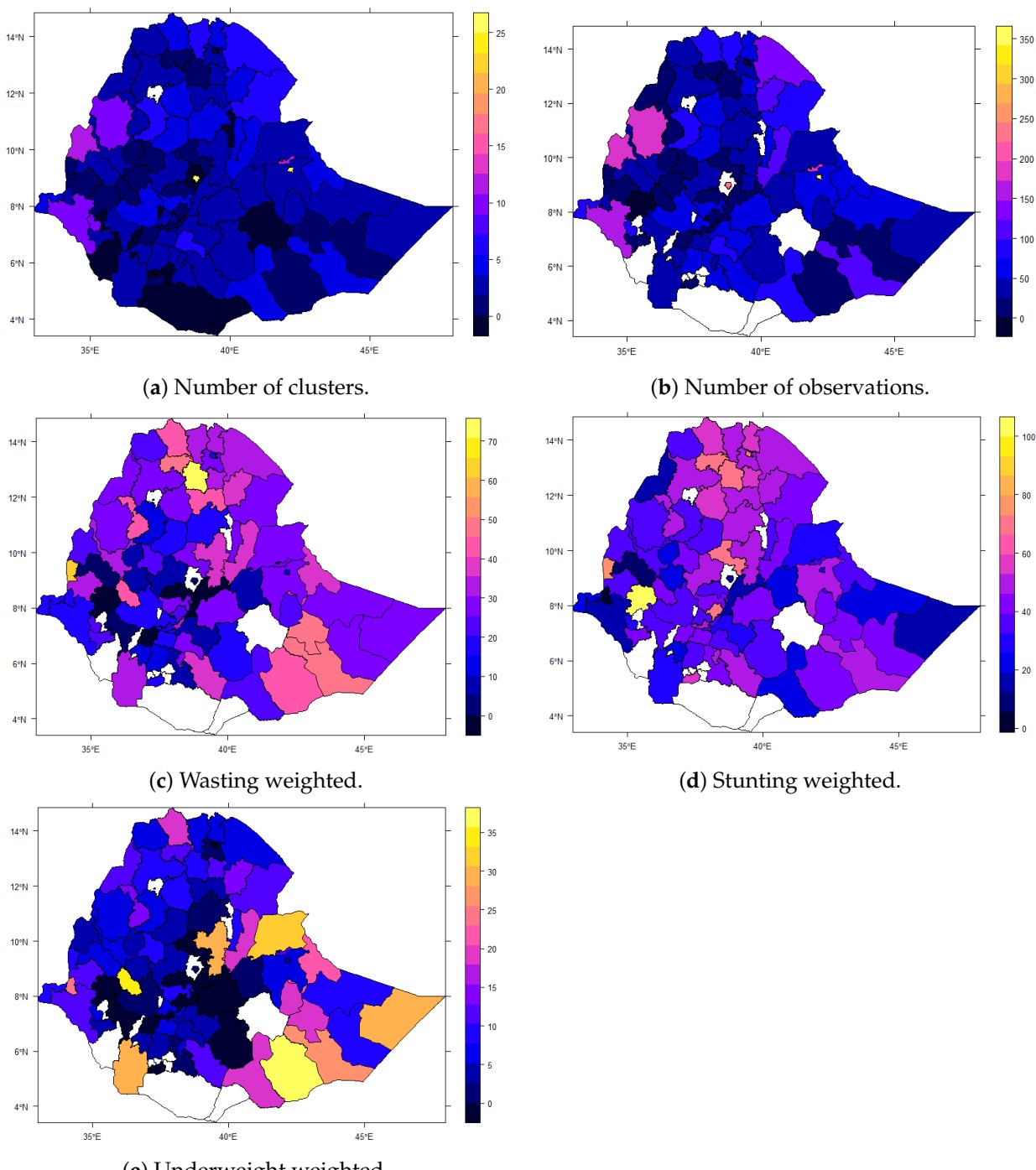

**Figure 2.** Number of clusters and weighted prevalence of malnutrition.

In general, we can see the spatial structure in the undernutrition estimates, namely in that two areas that are neighbors have more similar risks than two areas that are far apart. Specifically, spatial pattern analysis showed that there is spatial effect for being underweight, wasting, and stunting among children. The Global Moran's I test results are given in Table 2, showing the existence of significant spatial autocorrelation. Similarly, Anselin local Moran's I analysis and pattern (cluster and hotspot) analysis (Figure 3) showed the existence of significant clusters of a high as well as low prevalence of malnutrition. Hence, taking the spatial random effect in modeling evidently will have a great role. Therefore, the spatial random effect was taken into account in the small area estimation.

**Table 2.** Global Moran's I summary statistics for the malnutrition indicators.

| Measure | Underweight | Stunting | Wasting |
|---|---|---|---|
| Moran's Index | 0.218227 | 0.395968 | 0.372278 |
| Expected Index | −0.003289 | −0.003289 | −0.003289 |
| Variance | 0.002098 | 0.002122 | 0.002118 |
| z-score | 4.836414 | 8.667563 | 8.160307 |
| *p*-value | 0.000001 | 0.000000 | 0.000000 |

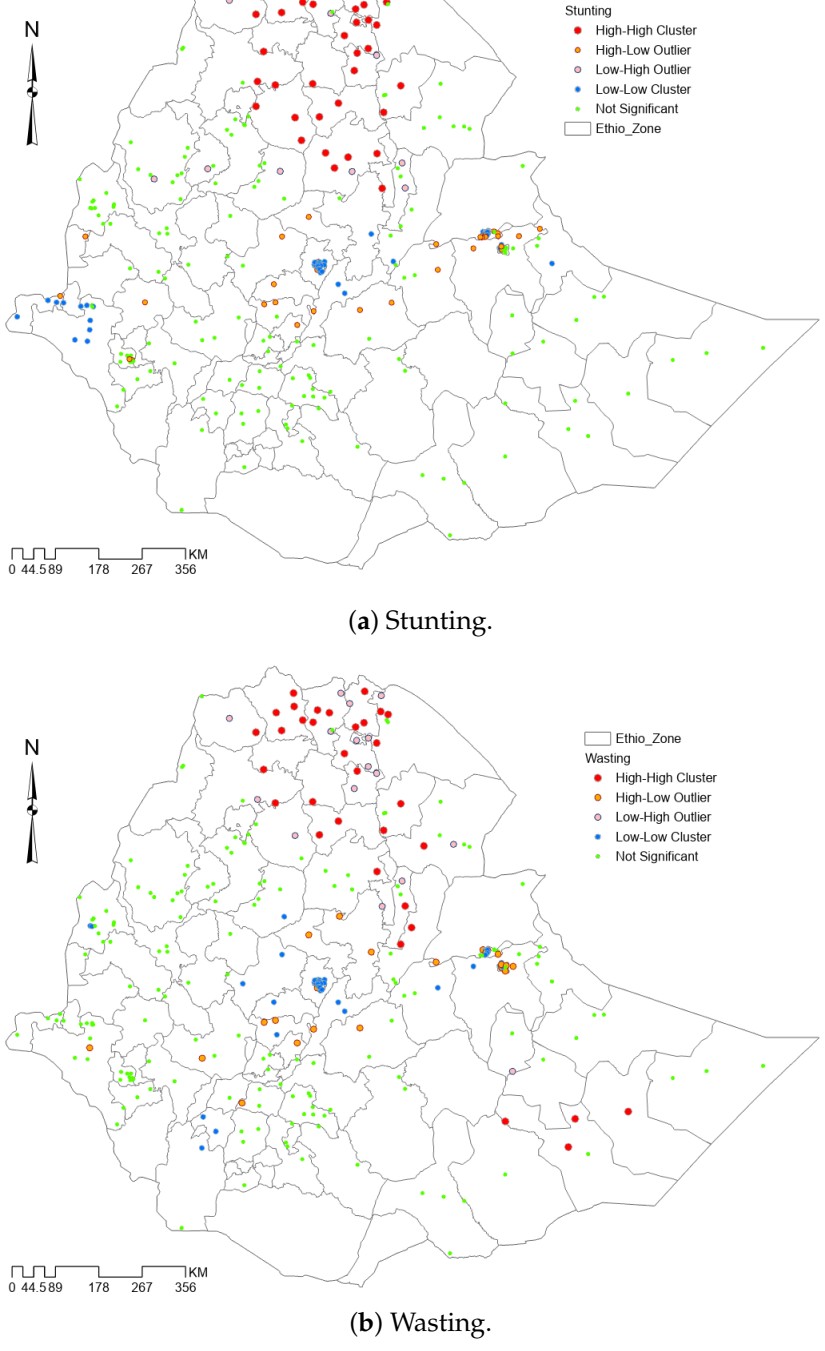

(**a**) Stunting.

(**b**) Wasting.

**Figure 3.** *Cont.*

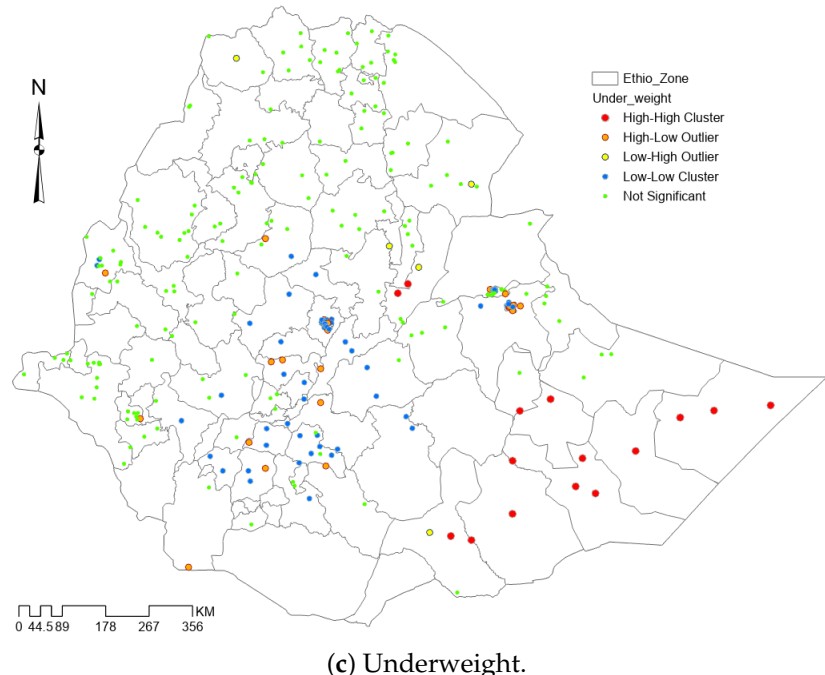

(**c**) Underweight.

**Figure 3.** Cluster and hotspot analysis of prevalence of malnutrition.

### 3.3. Small Area Estimates of Malnutrition

Some of the second administrative level in the country (zones) were not represented at all in the EMDHS 2019. These include the Oromiya Zone in Amhara, Yem Special, Sheka, Konta Special, Mirab Omo, Basketo, Alle, Derashe, Amaro and Burji in SNNP, Borena, East Bale in Oromia, and Daawa in Somali. Meanwhile, no woredas, the third adminstrative level in the country, in the unrepresented zones nor some more woredas from represented zones were represented at all, and a few of them had small sample sizes. Accordingly, the application of small area estimation is of paramount importance at the zonal level and woreda level. Aside from that, to comply with the assumption of the model, we considered the zone as secondary small area (SA2) and woreda as primary small area (SA1). The 2 resolutions with 72 knot points were considered for dimension reduction.

A total of 20,000 MCMC samples were generated from the posterior distribution. Considering a burnin period of 6000 and thinning for every third, 4667 samples were retained for the final process. The convergence and independence of the samples were confirmed from the trace plots (Figure 4), density plots (Figure 5), autocorrelation plots (Figure 6), and Geweke's test of convergence (Table 3). All these show that there is no evidence of assumption violations; that is, the samples were a realisation of stationary distribution.

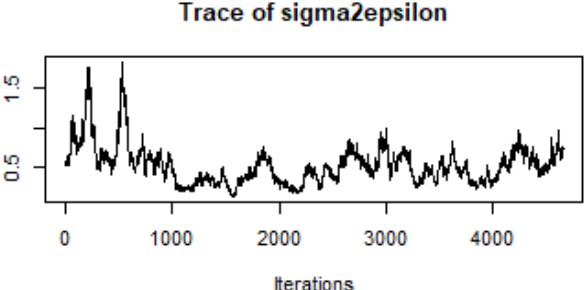

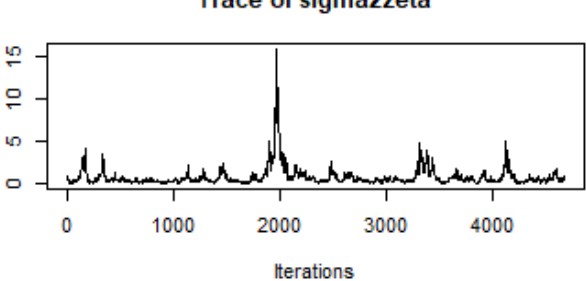

**Figure 4.** *Cont*.

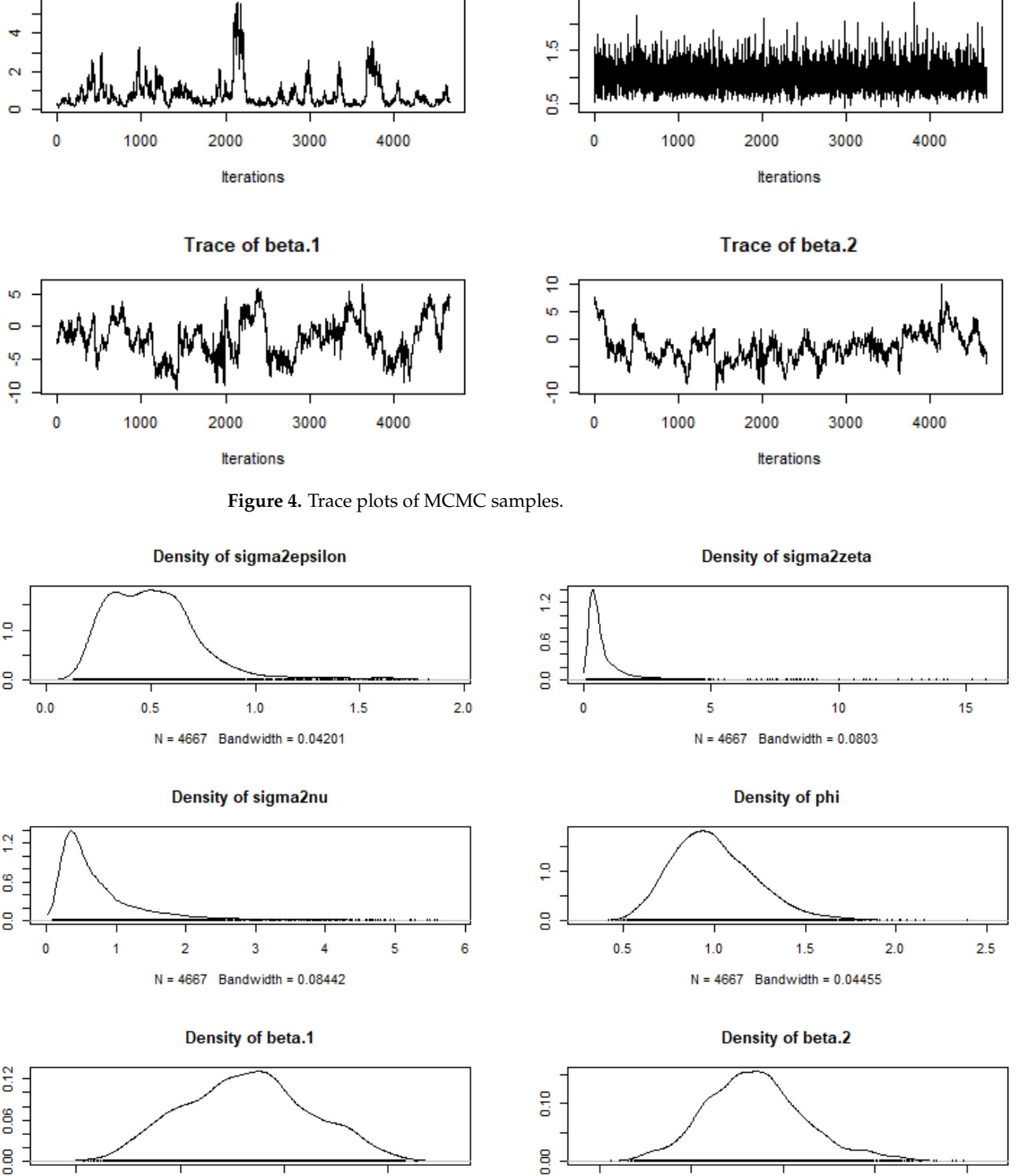

**Figure 4.** Trace plots of MCMC samples.

**Figure 5.** Kernel density plots of MCMC samples.

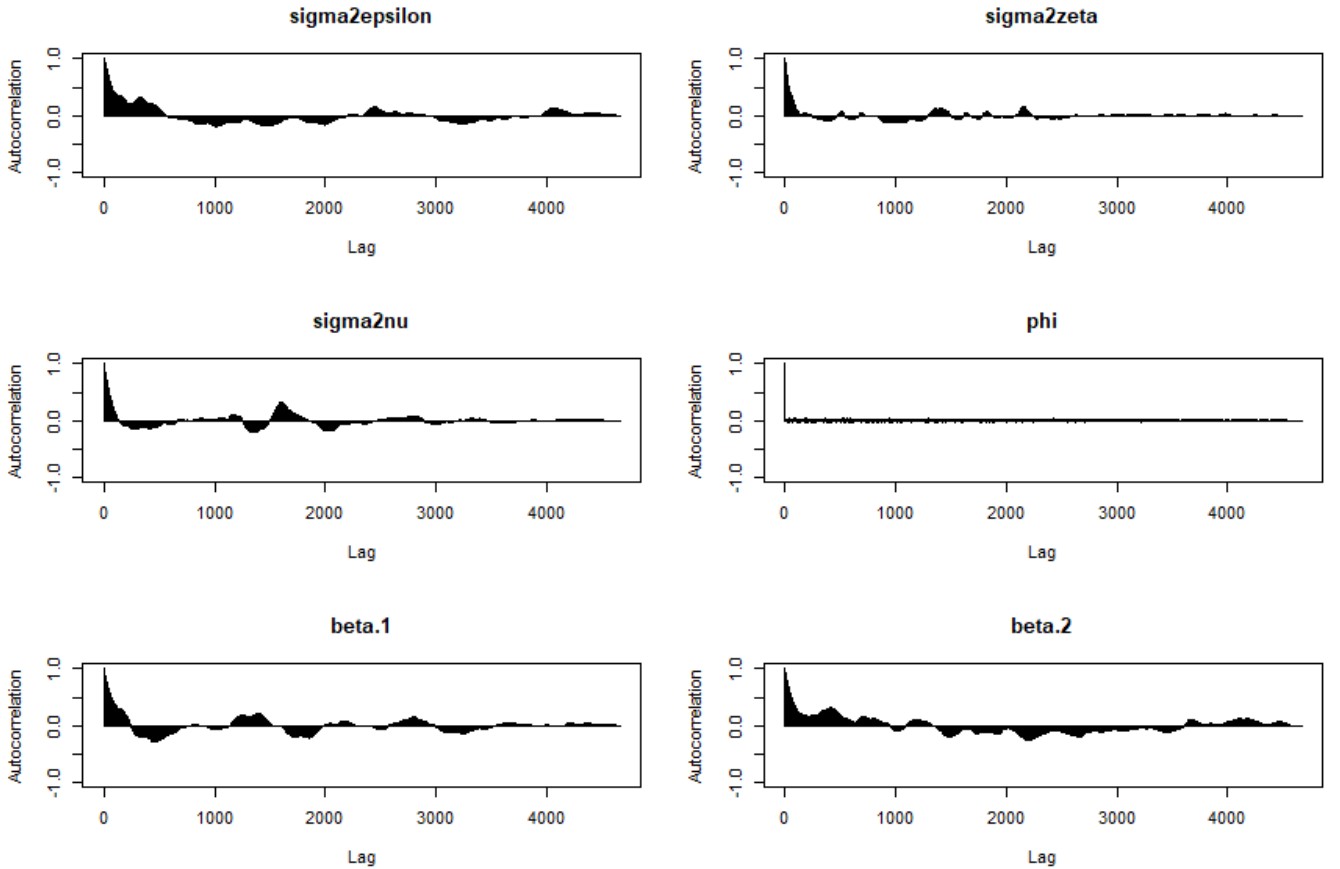

**Figure 6.** Autocorrelation function plots of MCMC samples.

**Table 3.** Geweke's test of convergence for application.

| Parameters | $\sigma_\epsilon^2$ | $\sigma_\zeta^2$ | $\sigma_\nu^2$ | $\phi$ | $\beta_1$ | $\beta_2$ |
|---|---|---|---|---|---|---|
| Z-value | 1.71403 | 0.32273 | 0.03037 | −0.05075 | 0.46410 | 0.19037 |

The PMSE plots were generated, showing the variability of the benchmarked HB estimates at the zonal level in Ethiopia (Figure 7).

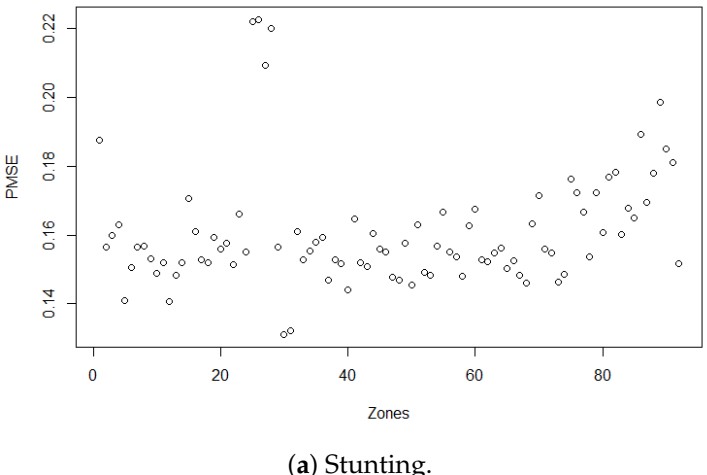

(**a**) Stunting.

**Figure 7.** *Cont.*

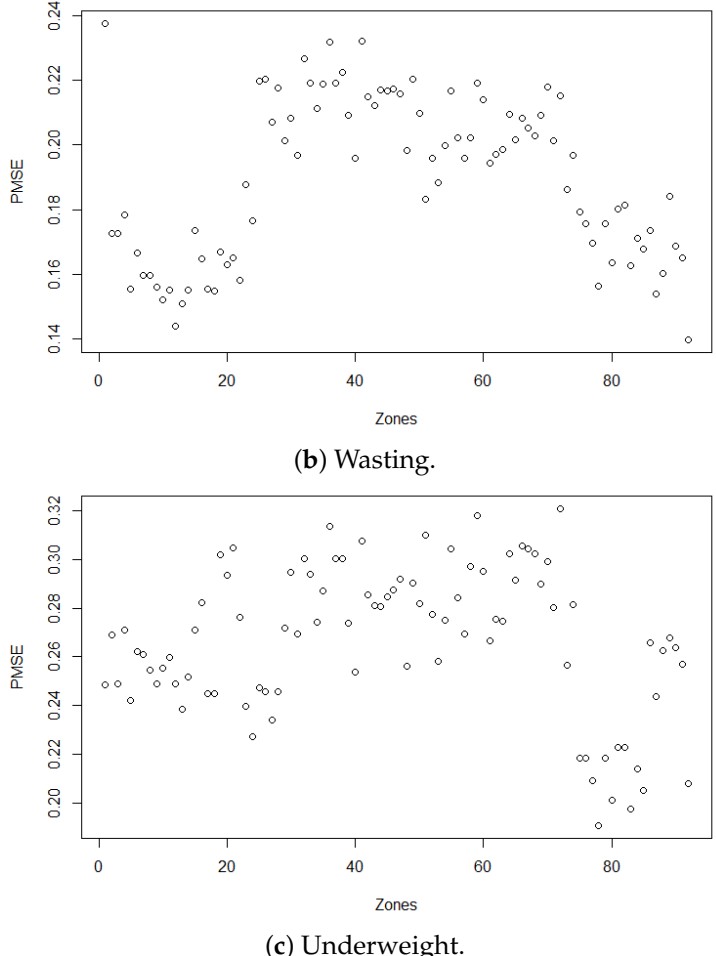

(**b**) Wasting.

(**c**) Underweight.

**Figure 7.** PMSE for benchmarked HB small area estimates.

The predictive posterior variance of prevalence of undernutrition and correction bias due to benchmarking are given in Figure 8.

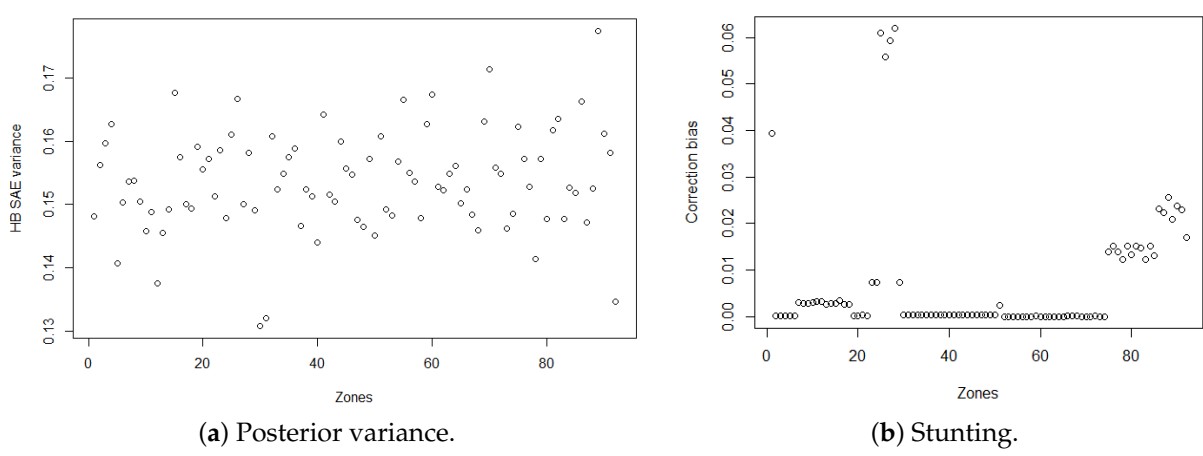

(**a**) Posterior variance.

(**b**) Stunting.

**Figure 8.** *Cont.*

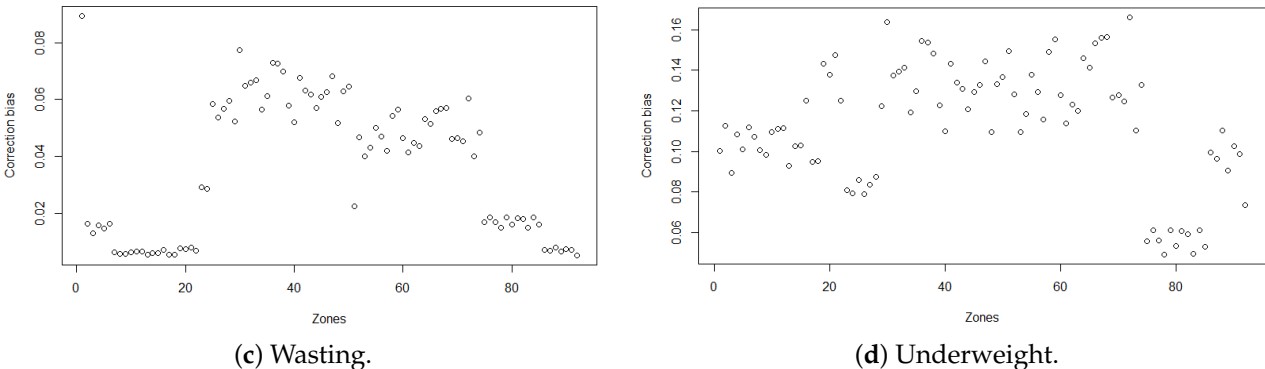

**Figure 8.** Posterior variance of HB SAE and benchmarking correction bias for undernutrition.

The estimates of malnutrition and its corresponding 95% credible interval using the proportion of literate persons and proportion of individuals from improved sources of drinking water as auxiliary variables are given. Accordingly, the plot for stunting generated from small area estimation is given in Figure 9. From the figure, we can understand the distribution of the burden of stunting among children under five at the zonal level in the country. The highest small area estimated prevalence of stunting was observed in the southwestern part of the country of Ethiopia. The corresponding measures of precision using the 95% credible interval is given in Figure 9, which shows the SAE-provided precise estimate.

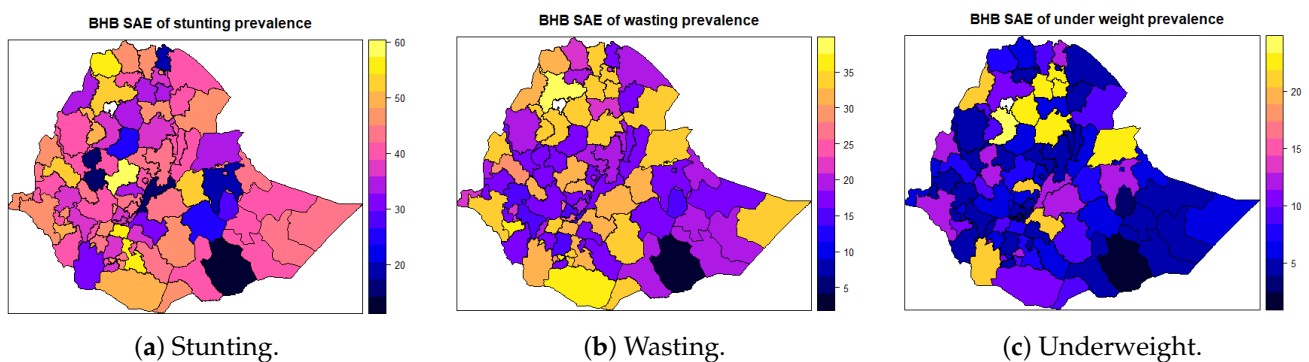

**Figure 9.** Prevalence of undernutrition from benchmarked HB small area estimation at zonal level in Ethiopia.

## 4. Discussion

The national level estimates of stunting (38.9%), wasting (22.5%), and being underweight (6.9%) indicate that stunting is still a severe public problem in the country, followed by wasting. This is in line with the related studies.

In addition to the national estimates, the local-level prevalence of malnutrition in Ethiopia was estimated. The design-based estimates were not adequate for estimation in the lower-level administrative areas, as the survey was not representative for lower-level administrative areas. There was a significant overall spatial autocorrelation, as well as hotspot areas with a high prevalence of undernutrition among children under five in Ethiopia. It was determined that including a spatial random effect in the estimation process was crucial in Ethiopia.

Accordingly, the hierarchical Bayesian spatial small area model was applied to estimate the prevalence of stunting, wasting, and being underweight. The estimate showed the existence of spatial variation of undernutrition at the zonal level.

In the midst of significant advancements in global economic growth, issues connected to child malnutrition have consistently posed a major challenge in low- and middle-income nations [2,31,32]. Globally, hunger and malnutrition diminish a country's gross domestic

product (GDP) by USD 1.4–2.1 trillion every year. Malnutrition costs the 54 African countries between 3 and 16 percent of their annual GDPs, with Ethiopia accounting for 16.5 percent, Malawi accounting for 10.3 percent, Rwanda accounting for 11.5 percent, and Burkina Faso accounting for 7.7 percent [3,31,33].

Increased agricultural productivity, girls' education promotion, immunisation, integrated management of neonatal and childhood illnesses, improved access to water and sanitation, and skilled birth delivery could all help to reduce the burden of undernutrition among Ethiopian children under the age of five [34].

The spatial variation of undernutrition at the zonal level could be related with economical variation, drought, food insecurity, and variation in cultivation [35]. This might suggest the need for the design and implementation of effective public health interventions at the zonal level to reduce undernutrition among children under five in Ethiopia.

This study is not without limitations. It is probable that there are other auxiliary variables that impact children's nutritional conditions, but the current study did not evaluate those variables due to a lack of information in the census data. Furthermore, the census data used were 15 years old, which may have influenced the results. Hence, readers are advised to take these limitations into account.

## 5. Conclusions

The prevalence of undernutrition among children under five in Ethiopia was estimated at the zone level using small area estimation techniques. The small area estimation concept was sought out by linking the most recent possible survey data and census data in Ethiopia. In Ethiopia, undernutrition had significant spatial variations across the country. The results specifically show that there is spatial variation in stunting, wasting, and being underweight across the zone level, showing that different locations faced different challenges and to different extents of undernutrition. Therefore, public health interventions that reduce undernutrition among children and enhance women's awareness toward undernutrition in zones with a high prevalence of undernutrition are crucial, and the Ethiopian Federal Ministry of Health (FMOH) should design tailored nutritional intervention for children under five who are living in zones with a high prevalence of undernutrition.

**Author Contributions:** Conceptualisation, K.F.M., A.K.W. and S.M.M.; methodology, K.F.M., A.K.W. and S.M.M.; software, K.F.M.; validation, K.F.M., A.K.W. and S.M.M.; formal analysis, K.F.M.; investigation, K.F.M.; resources, K.F.M.; data curation, K.F.M.; writing—original draft preparation, K.F.M.; writing—review and editing, K.F.M., A.K.W. and S.M.M.; visualisation, K.F.M.; supervision, A.K.W. and S.M.M.; project administration, K.F.M. All authors have read and agreed to the published version of the manuscript.

**Funding:** This research received no external funding.

**Institutional Review Board Statement:** The analysis presented in this study was based on the EMDHS 2019 and Census 2007, which are publicly available data sets with no identifiable information on the participants in the survey. Standard survey and census procedures for all ethical issues, like informed consent, were followed strictly in the EMDHS 2019 and Ethiopian Census 2007, respectively. Accordingly, for this study, no separate informed consent or ethical approval was required. However, we have received a grant of permission from Major DHS (http://dhsprogram.com, accessed on 5 February 2021) and IPUMS (https://international.ipums.org, accessed on 12 May 2021) to use survey data and census data, respectively.

**Data Availability Statement:** Minimal data that support the findings of this study can be accessed from the correspondence upon reasonable request. The data are not publicly available due to privacy or ethical restrictions.

**Acknowledgments:** We would like to express our appreciation to the Pan African University Institute of Basic Sciences, Technology and Innovation for the support. We also extend our appreciation to Major DHS and IPUMS for granting us permission to use the data. The authors wish to acknowledge the statistical office that provided the underlying data making this research possible: Central Statistical

Agency, Ethiopia. We also would like to acknowledge four anonymous reviewers for their insightful comments which helped us to improve the paper.

**Conflicts of Interest:** The authors declare no conflict of interest.

**Abbreviations**

ACF: autocorrelation Function; AIDS: acquired immunodeficiency syndrome; BHB: benchmarked hierarchical Bayesian; CSA: Central Statistics Agency; DHS: Demographic and Health Survey; EDHS: Ethiopia DHS; EMDHS: Ethiopian Mini DHS; EPHI: Ethiopian Public Health Institute; FMoH: Federal Ministry of Health; GDP: gross domestic product; HB: hierarchical Bayesian; HH: household; IPUMS: Integrated Public Use Microdata Series; MCMC: Markov chain Monte Carlo; PMSE: posterior mean squared error; RBHB: ratio-benchmarked hierarchical Bayesian; SA1: small area one; SA2: small area two; SAE: small area estimation; SD: standard deviation; SNNP: Southern Nations, Nationalities and People; SNNPR: Southern Nations Nationalities and People's Region; U5: under five; WHO: World Health Organization.

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
