# Peer review of "Small Area Estimation of Zone-Level Malnutrition among Children under Five in Ethiopia"

_mca, doi:10.3390/mca27030044_

Round 1

Reviewer 1 Report

In this paper, the authors conducted a statistical analysis on the prevalence of malnutrition among children under five at a detailed level in Ethiopia by analyzing the 2019 DHS survey and 2007 Census data. While the methods and techniques used in the analysis are routine, the outputs of the analysis are valuable addition to the literature, more importantly extremely valuable in its social value. Therefore, I recommend its publication with some minor revision. There is a minor concerns regarding the usage of 2007 census data for covariates. This might be the most recent census data available, it is still already about 15 years old. The authors should provide some comments and evidences that the covariates derived from this data set are still valid today. 

Author Response

Thank you for this important comment. As the reviewer acknowledged, the authors have no other choice but to use the most recent census, Census 2007. The benchmarking, included in this study to avoid model misspecification, may minimize issues arising related to validity of covariates from census with different time from the survey.  Furthermore, the statement “… the census data used was 15 years old, which may have influenced the results.” is included as a limitation in Page 13, Lines 348-349 of the manuscript.  

Reviewer 2 Report

The authors apply a modern statistical method to better support public health policy. The reason I accepted the offer to review this manuscript is because it is such an important application of statistical analysis.

Having said that, I must qualify my review. I have never used hierarchical Bayesian spatial modeling for small area estimation. The methods and results appear reasonable, but I lack the subject-matter expertise required for a rigorous review.

There are incomplete sentences in lines 210-211 and 288-289.  I found the sentences in the following lines to be confusing: 210-211, 222-223, 261-263, 291, 296-298, and 327.

Author Response

Thank you for the comments. The incomplete and confusing sentences are modified.  

Reviewer 3 Report

In this research, the authors employed small area estimation techniques to estimate the prevalence of malnutrition at zonal levels among under five children in Ethiopia. The small area estimation concept were sought by linking the most recent possible survey data and census data in Ethiopia. The result shows that there is spatial variation of stunting, wasting and underweight across zone level, showing different locations face different challenges and/or different extent.

The subject of the paper is interesting. I suggest the authors shorten the abstract and move the first part of that to the part introduction. The keywords are not suitable, for example Wasting cannot be keyword of this paper. Please revise the list of keywords. The math part of the paper is easy and correct. The novelty is not high but the results could be published.

Author Response

Thank you for the suggestion and the fruitful comments which can increase the visibility of the paper. The abstract is shortened as suggested. By referring the guidelines of the journal for authors and checking related papers, the keywords are updated as “Under Nutrition; Prevalence; Hierarchical Bayesian; Spatial Analysis; Small Area Estimation; Markov Chain Monte Carlo”. 

Reviewer 4 Report

My two main comments are:

1) The most important problem of the submitted paper is that in the SAE model Authors consider Census 2007 and survey 2019. There is an enourmous gap which has not been taken into account. There is a vast literature on that, mainly developed by World Bank related authors. I suggest to read examples on Albania reported in the edited book by Laderchi C.R., Savastano S. (eds.), Poverty and Exclusion in the Western Balkans, Economic Studies in Inequality, Social Exclusion and Well-Being 8, Springer Science+Business Media, New York.

2) There is no discussion at all on the choice of the SAE model adopted; in the presence of a census why Authors have not adopted the updated ELL ?

Author Response

  1. Thank you for this important comment. I have found the suggested document, specifically chapter 5, very important. Betti and his co-authors applied ELL which fills the gap between censuses by computing updated statistics.  The benchmarking, included in our study to avoid model misspecification, may minimize issues arising related to time difference between census and the survey. Furthermore, the statement “… the census data used was 15 years old, which may have influenced the results.” is included as a limitation in Page 13, Lines 348-349 of the manuscript.  
  2. Thank you dear reviewer for this comment which helped us to read about ELL. There might be a large number of different approaches of modeling a phenomenon, each with its own strength and limitations. We can see ELL is an approach that can be applied under the context of this study like the hierarchical Bayesian model used in this study. However, we thought this paper is more of an application study as a result we did not include detailed review on methodology.  

Round 2

Reviewer 3 Report

Now the paper is better and can be published.

Author Response

Dear Reviewer, I appreciate your efforts and the time you spent reviewing the manuscript.

Reviewer 4 Report

Authors have not taken into account my previous suggestions.

They have not responded to my main questions.

Author Response

Authors have not taken into account my previous suggestions. They have not responded to my main questions.

Reply: Dear Reviewer, Thank you for your time and comments. Please accept my apologies for misunderstanding your previous suggestions. We have attempted to reconsider your previous suggestions and have updated the manuscript accordingly. The previous main questions, along with their respective replies, are listed below. 

My two main comments are:

  • The most important problem of the submitted paper is that in the SAE model Authors consider Census 2007 and survey 2019. There is an enourmous gap which has not been taken into account. There is a vast literature on that, mainly developed by World Bank related authors. I suggest to read examples on Albania reported in the edited book by Laderchi C.R., Savastano S. (eds.), Poverty and Exclusion in the Western Balkans, Economic Studies in Inequality, Social Exclusion and Well-Being 8, Springer Science+Business Media, New York.

Reply: Thank you for your valuable input. We found the suggested document, specifically Chapter 5, very important and we have cited it in the paragraph (Page 6, Lines 216-231) briefing on the choice of the SAE model adopted.    

  • There is no discussion at all on the choice of the SAE model adopted; in the presence of a census why Authors have not adopted the updated ELL?

Reply: Dear Reviewer, Thank you for your helpful feedback on the manuscript. We have included a paragraph (Page 6, Lines 216-231) with a brief methodological review of the SAE model used. However, because we viewed this paper as more of an application study, the included review does not provide a thorough review of methodology.

Round 3

Reviewer 4 Report

Authors have eventually taken into account my comments/suggestions, even if in some cases they have not implemented theM. but at least they have reported the reasons in the paper and/or in the cover letter/answer.